# Adjuvant Therapy for Renal Cell Carcinoma: Hype or Hope?

**DOI:** 10.3390/ijms24044243

**Published:** 2023-02-20

**Authors:** Federica Cosso, Giandomenico Roviello, Gabriella Nesi, Sonia Shabani, Pietro Spatafora, Donata Villari, Martina Catalano

**Affiliations:** 1School of Human Health Sciences, University of Florence, 50134 Florence, Italy; 2Department of Health Sciences, Section of Clinical Pharmacology and Oncology, University of Florence, 50139 Florence, Italy; 3Department of Health Sciences, Section of Pathological Anatomy, University of Florence, 50139 Florence, Italy; 4Unit of Urological Robotic Surgery and Renal Transplantation, Careggi Teaching Hospital, 50134 Florence, Italy; 5Department of Experimental and Clinical Medicine, University of Florence, 50134 Florence, Italy

**Keywords:** adjuvant therapy, renal cell carcinoma, tyrosine kinase inhibitors, cancer immunotherapy, checkpoint inhibitors

## Abstract

Renal cell carcinoma (RCC) is the third most common genitourinary cancer accounting for approximately 180,000 deaths worldwide in 2020. Although over two-thirds of patients initially present localized disease, up to 50% of them may progress to metastatic disease. Adjuvant therapy aims to reduce the recurrence risk and improve outcomes in several types of cancers but is currently an unmet need in RCC. The results achieved with tyrosine kinase inhibitors in metastatic RCC led to the evaluation of these target therapies in an early setting with conflicting results for disease-free survival and no overall survival (OS) benefit. Likewise, the results of immune checkpoint inhibitors (ICIs) in an adjuvant setting are conflicting. Available data did not show an improvement in OS with ICIs in the early phase, although a positive trend for pembrolizumab has been recorded, receiving the Food and Drug Administration’s approval in this setting. However, the disappointing results of several ICIs and the heterogeneous pattern of RCC warrant biomarker identification and subgroup analyses to evaluate which patients could benefit from adjuvant therapy. In this review, we will discuss the rationale for adjuvant treatment in RCC, summarizing the results of the most important adjuvant therapy trials and current applications, to outline possible future directions.

## 1. Introduction

Renal cell carcinoma (RCC) counts approximately 400,000 diagnoses and 180,000 new deaths worldwide in 2020 [1]. It is among the top ten most common cancers in males, with a remarkable increase in incidence over the last few years [2]. Although most patients initially present localized disease, curable with complete nephrectomy, 20–50% of them will progress to metastatic cancer, while 25% to 30% of patients present with distant metastatic disease at diagnosis [3,4].

Over recent years, multiple tyrosine kinase inhibitors (TKIs) have been tested in patients with resected RCC, and all of these treatments showed an improvement in disease-free survival (DFS) or overall survival (OS) in the adjuvant setting, except for sunitinib, which showed only benefit in DFS [5,6,7,8,9,10].

The decision to recommend adjuvant treatment after nephrectomy in cancer disease is usually based on the relative risk of relapse, which is related to tumor characteristics and the patient’s ability to tolerate therapy. Recently, immunotherapy treatment with pembrolizumab has been shown to improve DFS in high-risk resected RCC patients, achieving Food and Drug Administration (FDA) approval [11].

In this review, we will discuss the rationale for adjuvant therapy in RCC, summarizing the results of the most important adjuvant therapy trials and current indications, to outline possible future directions.

## 2. Adjuvant Therapy in RCC

### 2.1. Rationale Use in RCC 

The United States National Cancer Database reports a 5-year (5 y) cancer-specific survival rate of 93% in patients with tumor–node–metastasis (TNM) stage I and II kidney cancers and of 71% and 14% for stage III /IV, respectively [12]. 

Generally, after surgical resection, most RCC recurrences occur within 3 years [13]; however, the authors showed that, after an initial 5 y disease-free interval, approximately 5% of patients experienced a renal recurrence and 15% a distant metastasis during the successive decade [14].

Overall, recurrence is reported in approximately 40% of surgically resected patients, and this relatively high risk suggests that many localized RCC patients may have micrometastatic disease at the time of nephrectomy [4]. Despite the initial nonmetastatic stage of RCC, the significant rate of patients who develop progression after nephrectomy supports the need for adjuvant therapy [15]. 

After curative resection, the intent of adjuvant therapy is to eradicate micrometastatic residual disease to reduce the risk of local and distant disease recurrence, with the goal of increasing the patient’s prognosis in terms of OS and DFS in patients at high risk of recurrence [16,17].

A fundamental aspect of adjuvant treatment is to identify patients at increased risk of recurrence who can benefit from adjuvant therapy in order to spare low-risk patients from the adverse effects of therapy [16]. 

### 2.2. Risk Classification 

The risk of recurrence is associated with specific disease characteristics at diagnosis. Primary tumor stage is a recognized prognostic factor, with up to 26% of patients with stage T2, about 50% of patients with stage T3, and nearly all patients with stage T4 disease reporting a recurrence after surgery [18,19]. The presence of sarcomatoid characteristics and higher tumor nuclear grade (G) are also independently associated with an increased risk of cancer recurrence [20,21]. 

Accordingly, in localized RCC, several validated prognostic and risk stratification models were designed for the post-surgery setting to assess the risk of relapse, such as the stage, size, grade, and necrosis staging system (SSIGN) score [22], the Leibovich scoring system, and the UCLA integrated staging system (UISS) [21,23]. In the UISS system, a score out of 180 was created to stratify patients into low-, intermediate-, and high-risk prognostic categories, combining symptoms, tumor size, histology, and pathological stage [23]. The authors clearly predicted OS and DFS regardless of histological subtype.

The use of the UISS certainly predicted 2 y and 5 y survival values irrespective of tumor histology in 76.5–86.3% of patients with non-metastatic disease, [23]. The Leibovich prognostic score was produced to evaluate the risk of developing metastatic disease; it integrates tumor size, stage, grade, histologic necrosis, and regional lymph node status into an algorithm [21] and stratifies patients into low-, intermediate-, or high-risk groups, accurately predicting metastasis-free survival. Both the UISS and Leibovich algorithms were validated, but the latter showed superior predictive accuracy [21,23]. These models are proven estimators of DFS, OS, and cancer-specific survival (CSS) and have been utilized in phase III trials of adjuvant therapy in RCC, ASSURE, and SORCE [5,7]. The SSIGN was created to predict CSS in patients with clear-cell (cc) RCC only [22]. 

### 2.3. Efficacy Outcome

In an adjuvant setting, OS is considered a gold-standard metric in clinical trials requiring studies of long duration. DFS and recurrence-free survival (RFS) are intermediate clinical endpoints and well-established surrogates for OS; both are FDA-sanctioned endpoints for colorectal and skin cancer in the adjuvant setting [24]. Considering that most patients with RCC who suffer metastatic disease die of cancer, these intermediate clinical endpoints may indicate the incidence of metastatic events and be considered a surrogate for OS in RCC as well [24].

In their meta-analysis, Harshman et al. evaluated DFS as an early clinical surrogate for OS in the adjuvant setting for localized RCC. Thirteen randomized clinical trials (RCTs) enrolling 6473 patients have been evaluated, showing only a modest relationship between 5 y DFS and 5 y OS rates (R-squared, 0.48; 95% confidence interval (CI) 0.14–0.67) and between treatment effects measured by DFS and OS hazard ratios (HRs) (R-squared, 0.44; 95% CI 0.00–0.69) [24]. 

A recent retrospective observational study proved a statistically significant correlation between DFS and OS (95% CI: 0.65–0.74; *p* < 0.001) in 643 newly diagnosed RCC patients completely resected. Recurrence in patients with intermediate–high- or high-risk RCC was related to a significantly shorter OS, resulting in a concrete positive association between DFS and OS in this population [25].

## 3. Adjuvant Tyrosine Kinase Inhibitors 

In the last few years, targeted therapy, such as vascular endothelial growth factor receptor (VEGFR)-TKIs, have significantly changed the process of treatment in patients with metastatic RCC, providing impetus for clinical trials aimed at recognizing an effective adjuvant treatment [26]. Globally, five phase III randomized controlled trials were performed, yielding conflicting results for DFS; no OS advantage was identified in any of these studies [27].

The ASSURE study evaluated adjuvant sunitinib or sorafenib in comparison with placebo in 1943 patients with RCC at intermediate or high risk of recurrence [7]. The treatment regimen required a dose adjustment due to toxicity issues for both sunitinib (from 50 to 37.5 mg) and sorafenib (from 400 mg to 200 mg twice daily). Differences in DFS or OS outcomes were not revealed.

The S-TRAC of 615 patients with high-risk recurrent RCC after nephrectomy [9]: Individuals were randomized 1:1 to sunitinib 50 mg once a day, four weeks on treatment followed by two weeks off, or placebo for 1 year. The median DFS (mDFS) for sunitinib and placebo was 6.8 versus 5.6 years (HR: 0.76, 95% CI: 0.59–0.98, *p* = 0.03). Based on the significant difference achieved in DFS, although median OS was not reached, sunitinib received, in November 2017, FDA approval for adjuvant therapy in high-risk adult patients. 

Afterward, sorafenib failed to demonstrate differences in DFS and duration of OS vs. placebo in the SORCE study [5]. This trial included patients at intermediate-risk or high-risk ccRCC or non-ccRCC (as per the Leibovich risk model) randomizing to sorafenib or placebo for a treatment period of 3 years. The ten-year OS rate was 69% for placebo, 70% for 3 years of sorafenib, and 69% for 1 year of sorafenib [5].

The PROTECT trial, evaluating adjuvant pazopanib in patients with locally advanced RCC at high risk of recurrence after surgery, failed the primary endpoint DFS in the intention-to-treat population (HR, 0.86; 95% CI, 0.70–1.06; *p* = 0.165) [8]. There was no significant difference in OS between the pazopanib and placebo arms [28].

The ATLAS trial included 724 patients who had undergone nephrectomy having >50% ccRCC. Subjects received oral axitinib 5 mg or placebo for ≤3 years twice daily [6]. Axitinib did not achieve the primary endpoint DFS (HR 0.870; 95% CI: 0.660–1.147; *p* = 0.3211) vs. placebo. 

In the latest EVEREST study, everolimus 10 mg once a day for 54 weeks was evaluated in 2018 patients with kidney cancer at intermediate–high or very high risk of relapse who underwent nephrectomy [10]. An improved RFS was assessed in patients who received nine courses of 6 weeks of everolimus vs placebo (HR 0.85, 95% CI, 0.72–1.00; P1-sided = 0.0246). OS was similar between treatment arms (HR 0.90, 95% CI, 0.71–1.13: P1-sided = 0.178. Overall, VEGFR-TKIs were not demonstrated to improve survival outcomes in the adjuvant setting [27] (Table 1).

## 4. Rational Use of Immunotherapy in an Adjuvant Setting

The debatable clinical benefit of adjuvant TKI therapy at the cost of high adverse events incidence has led the way for the development of different perioperative strategies, such as immune checkpoint inhibitors (ICIs) designed to reinforce and enhance immune activity against cancer cells.

Monoclonal antibodies against immune checkpoints such as programmed death-1/-ligand 1 (PD-1/L1) and cytotoxic T-lymphocyte-associated protein 4 (CTLA-4) have shown impressive efficacy in the metastatic setting of RCC, changing the first-line therapeutic algorithm [29,30].

PD-L1 is expressed by immune cells under inflammatory conditions and by tumor cells as an “adaptive immune mechanism” to escape anti-tumor responses; it induces an inhibitory response to T cells by binding PD-1 expressed on T cells, generating immune tolerance toward tumor cells. PD-1 and PDL-1 checkpoint inhibitors selectively block the interaction between PD-1 and PD L-1, potentially restoring effective antitumor immunity [16]. Cytotoxic T-lymphocyte-associated protein 4 (CTLA-4) is a checkpoint receptor on cytotoxic lymphocytes that ligates with B-7 exhibited on antigen-presenting cells and suppresses T-cell proliferation [16] (Figure 1).

Several pieces of evidence suggest that primary tumor surgical resection destroys the host’s immune system [31]. These effects lie within the “postoperative period”, which lasts days to weeks after tumor surgical resection, potentially creating an immunosuppressive window for the expansion and escape of occult tumors [31].

In preclinical and clinical metastatic settings, ICIs against PD-1 have been demonstrated to relieve postoperative T-cell dysfunction while increased T-cell activation has been shown following CTLA-4 inhibitor administration [31].

Thanks to a durable response rate and a manageable safety profile, ICIs have generated enthusiasm for the potential utility of such therapies in the adjuvant setting, as well as the maintenance of antitumor efficacy, even after their discontinuation [16,32,33]. 

### Adjuvant Immune Checkpoint Inhibitors

KEYNOTE-564 [34] is a randomized phase III trial that evaluated pembrolizumab vs. placebo for one year in intermediate–high-risk pT2-pT3 N0, M0 RCC or high-risk (pT4, N0, M0, or any pT, any grade, N+, M0) or stage M1 with no evidence of disease (M1 NED) [34]. No prior systemic treatment for advanced RCC and disease-free survival after surgery were required. The primary endpoint was DFS by investigator assessment (INV-DFS), described as the time from randomization to the first documented relapse of RCC, secondary systemic malignancy, or death from any cause; whichever occurred first. At the first interim analysis after 30 months, the DFS was met with an HR of 0.63 (95% CI 0.50–0.80 *p* < 0.0001). OS was superior for the pembrolizumab arm in comparison with the placebo (HR 0.52, 95% CI 0.31–0.86, *p* = 0.0048). The most common adverse events (AEs) were hypertension and increased alanine aminotransferase.

In 2021, the FDA approved pembrolizumab in an adjuvant setting, which was certainly a milestone in the history of RCC treatment [35].

IMmotion010 is an ongoing randomized phase III trial aimed to assess atezolizumab, a PD-L1 inhibitor, vs placebo, for one year, in the setting of high-risk ccRCC or sarcomatoid (T2 grade 4, T3a grade 3–4, T3b/c any grade, T4 any grade, or Tx N+ any grade) after local surgery, M0, or M1NED [36]. Atezolizumab did not improve clinical outcomes in the ITT population [36]. The primary endpoint was the INV-DFS at a median follow-up of 44.7 months, which resulted in 57.2 months with atezolizumab and 49.5 months with placebo (HR 0.93, 95% CI 0.75–1.15; *p* = 0.50). OS was not evaluable (HR 0.97, 95% CI 0.67–1.42) [37]. Grade 3 or 4 AEs occurred in 27% and 21% of patients receiving atezolizumab or placebo, respectively. 

A further ongoing ICI-adjuvant RCC trial is the Checkmate-914 phase III study, which analyzes nivolumab, an anti-PD-1 monoclonal antibody, alone or combined with ipilimumab, an anti-CTLA4 monoclonal antibody, vs placebo for 24 weeks in high-risk mostly ccRCC with pT2a (grade 3 or 4), pT2b/pT3/pT4 (any grade), N0, or any T (any grade) N1 after complete or partial nephrectomy [38]. Checkmate-914 failed the primary efficacy endpoint of INV-DFS for nivolumab with ipilimumab vs placebo (HR 0.92, 95% CI 0.71–1.19; *p* = 0.5347). Grade ≥ 3 treatment-related AEs were described in 28.5% vs. 2.0%, respectively. Data on nivolumab monotherapy are not currently available. The safety profile of nivolumab plus ipilimumab was in line with its known profile in advanced RCC, although a higher rate of discontinuation due to treatment-related AEs was reported with the combination vs placebo in this trial [38].

The PROSPER trial was a randomized phase III study that evaluated nivolumab in a perioperative setting for stage T2 or greater or lymph-node-positive M0 RCC of any histology [39]. Pre-surgery participants were randomized to receive two doses of neoadjuvant nivolumab succeeded by adjuvant nivolumab for 9 months or standard nephrectomy followed by observation. The trial was stopped early due to futility. The primary endpoint was RFS, which reported similar results between the two arms; OS was not mature at the time of analysis but was not statistically different between study arms. The 20% of patients treated with nivolumab experienced at least one grade 3–4 AE, compared with 6% in the control arm. Another randomized phase III trial involving ICIs is the RAMPART study, which is evaluating durvalumab, an anti-PD-L1 monoclonal antibody, plus tremelimumab, an anti-CTLA-4 monoclonal antibody, in post-surgery intermediate–high-risk RCC [40]. The primary endpoints are DFS and OS. The planned enrolment is 1750 patients.

Finally, the LITESPARK 002 was designed to compare the efficacy and safety of belzutifan, HIF-2α inhibitor, in combination with pembrolizumab vs placebo, plus pembrolizumab as an adjuvant treatment of ccRCC at intermediate–high or high risk, including M1 NED [41]. The primary endpoint is DFS, and the planned enrolment is 1600 patients [42]. The data are summarized in Table 2.

## 5. Current Challenges and Future Perspective 

Surgery remains the standard of care for non-metastatic RCC. The significant rate of patients who develop progression after nephrectomy, approximately 40%, justifies the need for adjuvant therapies in RCC with a high risk of relapse [4,15].

Several existing risk models were conceived to identify the high-risk population, including not only tumor characteristics but also patient features such as ECOG performance status [43]. The most widely used models are the SSIGN score [22] and the Leibovich and the UISS scoring systems [21,23], as previously mentioned, which have been incorporated into eligibility criteria for previous adjuvant studies. Although they are well-known algorithms created for supporting physicians to classify patients into low-, intermediate-, or high-risk groups, they rely on retrospective and historical data spanning decades and are not personalized to the individual patient. A great variability rate of occurrence within each risk group has been detected [43]. 

In patients with a high risk of recurrence, an effective adjuvant therapy needed to be detected, and the introduction of targeted therapies and the significant results demonstrated in the metastatic setting generated great enthusiasm for the possibility of expanding their use in the non-metastatic RCC as well.

In recent years VEGFR-TKIs and IO have significantly improved survival outcomes for patients with mRCC [26]. Nevertheless, in the adjuvant RCC setting, these treatments yielded conflicting results across VEGFR-TKI randomized phase III trials [27].

Thanks to the DFS results of the S-TRAC study [9] sunitinib received FDA approval for adjuvant treatment in high-risk adult patients, although no other study including VEGFR-TKIs achieved a DFS advantage vs placebo. Moreover, no OS benefit was observed in any of the VEGFR-TKI studies [27]. 

Promising results showing that immunotherapies are more effective than TKIs in metastatic disease created good interest to evaluate whether this could be translated into adjuvant RCC [26].

The latest results from KEYNOTE-564 showed that pembrolizumab could be a new standard of care for adjuvant therapy in patients with ccRCC and its approval was a landmark for designing the IMmotion010 and the CheckMate 914 studies, despite failing to achieve an improvement in their primary outcome of DFS [34,35,36,38]. The conflicting findings may be due to distinct differences in eligibility criteria for the ongoing ICI adjuvant trials, either as a single agent or combined with anti-CTLA4 [44].

The RECUR dataset investigated how differences in eligibility criteria may impact study results in RCC adjuvant trials [32]. It estimated the variability in DFS and OS benchmarks of the placebo groups for the ongoing adjuvant immune-oncology trials. The results suggested an extended DFS and OS for the placebo group based on the current eligibility criteria for the CheckMate-914, PROSPER, and RAMPART studies, with a lower event rate for DFS than for the IMmotion10 and KEYNOTE-564 studies [34,36,38,39,40]. 

These data could be explained by the study population contained in IMmotion010 and KEYNOTE-564 trials, which allowed the inclusion of resected M1 disease (M1 NED) [34,36]. In the KEYNOTE-564 subgroup analysis, a numerically much higher benefit for pembrolizumab-treated M1 patients was shown in terms of DFS (HR 0.28, 0.12–0.66) than was observed for M0 intermediate–high risk (HR 0.68, 0.52–0.89) or M0 high risk (HR 0.60, 0.33–1.10) [34]. Nevertheless, in the IMmotion010 trial, equal results were not achieved for M1 NED patients treated with atezolizumab (HR 0.93, 0.58–1.49), in comparison with M0 intermediate–high risk (HR 0.93, 0.69–1.25) and M0 high risk (HR 0.93, 0.61–1.41) [45].

The conflicting results obtained within these subgroup analyses should be interpreted with caution, also considering the small sample size included.

The currently recruiting phase III trials RAMPART and LITESPARK 002 plan to include more than 3000 patients [40,41]. Percentages of eligible patients vary owing to heterogeneity in trial eligibility criteria [32]. The RAMPART trial includes patients with Leibovich risk scores of 3 to 5, although limited to 25% of the trial population, which may result in a very long DFS with an extended follow-up [40]. This trial may provide interesting information, as it is recruiting patients with all cell types of RCC, except for pure oncocytoma, collecting duct, medullary, and transitional cell cancer [40].

The LITESPARK 002 trial plans to include patients who report an intermediate–high, high-risk, and M1 NED [41]. A subgroup analysis could provide more information on patients undergoing metastasectomy after nephrectomy.

The results confirm that the most appropriate intermediate–high-risk patient group for adjuvant management requires carefully designed clinical trials [44]. 

To date, only KEYNOTE-564 [34] has described a significant DFS benefit for immunotherapy in an adjuvant setting with an improved trend in OS. Unfortunately, OS data are still immature, and a statistical OS benefit has not been proven. Likewise, OS data are currently immature for all IO trials in this setting. As we await OS results, in a recent retrospective analysis of the SEER database, Kendall’s τ rank relation between DFS and OS was shown, suggesting that DFS is a reasonable surrogate endpoint while OS is immature [46].

The next generation of markers might produce personalized prognostic information and predict responses to systemic treatments [44]. Several molecular variables have been investigated; Klatte et al. [47] demonstrated that Ki-67, p53, endothelial VEGFR-1, epithelial VEGFR-1, and epithelial VEGF-D may predict DFS in non-metastatic ccRCC. Thompson et al. analyzed 196 clinical specimens, revealing that an increased expression of the B7 family members (B7H1), which are involved in the immune response, is linked to poorer prognosis in RCC [48]. The role of circulating tumor cells in RCC and their possible utility in prognostication remain unclear [44]. 

A further crucial key point for adjuvant therapies is the safety profile and the associated risk of discontinuation due to AEs. The related toxicity and side effects played a central role in specifically targeted therapy trials as adjuvant treatment [15]. The ASSURE trial required a starting dose reduction due to toxicities; however, the number of patients who reported a high-grade AE in those who started at a reduced dosage still exceeded 55% in both the sunitinib and sorafenib groups [7], with an overall discontinuation rate of 34% and 30%, respectively. Likewise, the PROTECT study needed an amendment of the primary objective due to toxicity in a cohort that randomly received a reduced dose of pazopanib (from 800 mg to 600 mg) [8]. However, similar discontinuation rates for the 600 mg and 800 mg groups of pazopanib were described because of adverse effects (35% and 39%, respectively). Moreover, in the EVEREST trial, 37% of patients withdrew in the everolimus treatment arm due to AEs, compared to 5% in the placebo group [10]. 

The safety of pembrolizumab, atezolizumab, and nivolumab/ipilimumab reported in clinical trials was consistent with their known profile in advanced RCC [34,36,38]. 

Nonetheless, the treatment-related AEs leading to discontinuation were much higher for the active treatment arm vs. placebo in both the KEYNOTE-564 trial (21% of pembrolizumab vs. 2.2% of placebo) and Checkmate 914 trial (29.0% of nivolumab plus ipilimumab vs. 1.0% of placebo) [34,38]. In the IMmotion010 trial, atezolizumab reported treatment-related AEs that led to discontinuation, like the placebo (11.5% vs. 2.6%, respectively) [36].

In particular, the nivolumab plus ipilimumab safety profile analysis showed that any-grade treatment-related AEs were reported in 88.9% of patients vs. 56.8% of the placebo treatment group; grade ≥ 3 treatment-related AEs were reported in 28.5% vs. 2.0%, respectively [38].

Although the occurrence of severe immunotherapy-related AEs could require adjuvant therapy suspensions, studies are needed to better guide therapy resumption in these patients [32]. In conclusion, the heterogeneous pattern of RCC and the negative results for several ICIs warrant further follow-up, biomarker identification, and subgroup analyses to identify which patients could derive the most advantage from adjuvant therapy with ICIs. Meanwhile, a careful discussion of the risk–benefit ratio of adjuvant treatment after surgery for RCC is needed, and treatment decisions should be made with attention to each patient to avoid overtreatment and toxicities. 

## Figures and Tables

**Figure 1 ijms-24-04243-f001:**
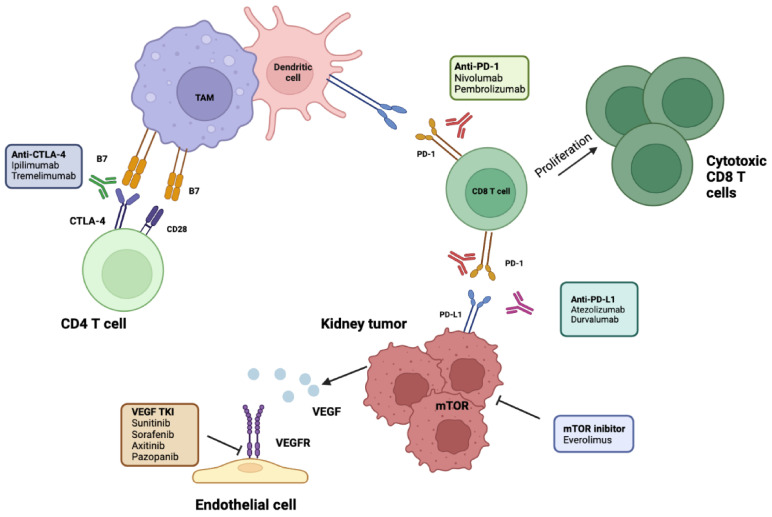
Molecular and immunotherapy targets evaluated in adjuvant RCC. ICIs of PD-1 disrupt its interaction with PD-L1, leading to enhanced T-cell proliferation and activation. Antibodies targeting PD-L1 prevent its interaction with PD1 on CD8 T cells, allowing their activation. Anti-CTLA-4 antibodies allow CD28 to bind to its receptor, B7, and activate naïve CD4 T cells. Antiangiogenetic targets include TKIs on VEGFR. mTOR inhibitors prevent tumor growth. TAM, tumor-associated macrophage; RCC, renal cell carcinoma; ICIs, immune checkpoint inhibitors; PD-1/L-1, programmed cell death 1/ligand-1; CTLA4, cytotoxic T-lymphocyte antigen 4; TKIs, tyrosine kinase inhibitors; VEGFR, vascular endothelial growth factor receptor, mTOR, mammalian target of rapamycin; TAM, tumor-associated macrophages. Created in BioRender.com.

**Table 1 ijms-24-04243-t001:** Clinical trials on tyrosine kinase inhibitors in RCC adjuvant treatment.

Clinical Trial [Ref.]	No. of Patients	Population	Treatment Arms (*n*)	Duration of Treatment	Primary Endpoint	Results	Overall Response	Grade 3 or Worse Adverse Events
ASSURE [7]	1069	≥T1b completely resected, M0	Sunitinib 50 mg (*n* = 358) Vs. Sorafenib 400 mg(*n* = 355) Vs. Placebo (*n* = 356)	1 year	DFS	HR, 1.02 (97.5% CI 0.85–1.23; *p* = 0.8038) HR, 0.97 (97.5% CI, 0.80–1.17; *p* = 0.7184)	5-y-OS: HR, 1.06; 97.5% CI, 0.78–1.45; *p* = 0.66; HR, 0.80; 97.5% CI, 0.58–1.11; *p* = 0.12	Sunitinib: 63% Sorafenib: 72% Placebo: 25%
S-TRAC [9]	615	T3-T4, N0-Nx, M0 or any T, N+, M0	Sunitinib 50 mg (*n* = 309) Vs. placebo (*n* = 306)	1 year	DFS	HR, 0.76; 95% CI, 0.59 to 0.98; *p* = 0.03	mOS HR 0.92, 95% CI, 0.66–1.28; *p* = 0.6)	Sunitinib: 65% Placebo: 23.3%
PROTECT [8]	1538	pT2 high grade, pT3-4, N0/+, M0	Pazopanib 600 mg (*n* = 769) Vs. Placebo (*n* = 769)	1 year	DFS	HR, 0.86; 95% CI, 0.70 to 1.06. *p* = 0.165	mOS HR, 1.0, 95% CI, 0.80–1.26; *p* > 0.9	Pazopanib: 59% Placebo: 19%
ATLAS [6]	724	>pT2 and/or N+ M0	Axitinib 10 mg (*n* = 363) Vs. Placebo (*n* = 361)	1 to 3 years	DFS	HR = 0.870; 95% CI: 0.660 to 1.147; *p* = 0.3211	NA	Axitinib: 49% Placebo:12%
SORCE [5]	1711	Intermediate- or high-risk disease (Leibovich score 3 to 11)	Sorafenib for 1-year followed by 2-year placebo, (*n* = 642) Vs. 3-year sorafenib (*n* = 639) Vs. Placebo (*n* = 430)	3 years	DFS	HR, 1.01; 95% CI, 0.82 to 1.23. *P* = 0.946)	HR, 0.92; 95% CI, 0.71 to 1.20; *p* = 0.541 HR, 1.06; 95% CI, 0.82 to 1.38; *p* = 0.638	Sorafenib plus placebo: 59% Sorefenib: 64% Placebo: 29%
EVEREST [10]	2018	pT1-pT3a N0, pT3-pT4, N0/+, M0	Everolimus 10 mg (*n* = 775) Vs. Placebo (*n* = 770)	5 years	RFS	HR 0.85, 95% CI, 0.72–1.00; P1-sided = 0.0246	HR 0.90, 95% CI, 0.71–1.13: P1-sided = 0.178	Everolimus: 46% Placebo: 11%

DFS, disease-free survival; RFS, relapse-free survival; mOS, median overall survival; HR, hazard ratio; CI, confidence interval.

**Table 2 ijms-24-04243-t002:** Clinical trials on immune checkpoint inhibitors in RCC adjuvant treatment.

Clinical Trial [Ref.]	No. of Patients	Tumor Features	Treatment Arms	Duration of Treatment	DFS	RFS	OS	Grade 3 or Worse AEs
KEYNOTE-564 [34]	994	Intermediate–high-risk M0-M1 NED Clear-cell RCC/sarcomatoid	Pembrolizumab Placebo	1 year	HR 0.63 (95% CI 0.50–0.80 *p* < 0.0001)	75.2% (95% CI 70.8–79.1) 65·5% (60.9–69.7)	HR 0.52 (95% CI 0.31–0.86, *p* = 0.0048)	32% 18%
IMmotion010 [36]	778	Intermediate–high-risk M0-M1 NED Clear-cell RCC/sarcomatoid	Atezolizumab Placebo	1 year	HR 0.93 (95% CI 0.75–1.15, *p* = 0.50)	NA	HR 0.97 (95% CI 0.67–1.42)	28% 24%
Checkmate-914 [38]	816	Intermediate-high risk M0 Clear-cell RCC/sarcomatoid	Nivolumab + Ipilimumab Placebo	At least 24 weeks	HR 0.92 (95% CI 0.71–1.20)	NA	NA	28.5% 2%
PROSPER [39]	819	Intermediate-high risk, M0 or M1 NED RCC of any histology	Nivolumab neoadjuvant- adjuvant Placebo	40 weeks (One dose prior to surgery followed by 9 doses)	NA	HR: 0.97 (95% CI: 0.74–1.28; P1-sided = 0.43)	HR: 1.48; (95% CI: 0.89–2.48; P1-sided = 0.93).	20% 6%
LITESPARK 002 [41]	1600	Clear-cell RCC pT2, grade 4 or sarcomatoid, N0, M0 or pT3, any grade, N0, M0, high (pT4, any grade, N0, M0 or pT, any stage/grade, N+, M0) or M1 NED	Belzutifan + pembrolizumab Placebo + pembrolizumab	1 year	DFS	NA	NA	NA
RAMPART [40]	1750	Clear-cell and non-clear-cell histological RCC subtypes with high or intermediate risk of relapse (Leibovich score 3–11).	Durvaumab + tremelimumab Placebo	1 year	DFS and OS	NA	NA	NA

NED, no evidence of disease; RCC, renal cell carcinoma; DFS, disease-free survival; RFS, relapse-free survival; CI, confidence interval; AEs, adverse events; OS, overall survival; NA, not available.

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
