# Peer review of "Adjuvant Therapy for Renal Cell Carcinoma: Hype or Hope?"

_ijms, 2023, doi:10.3390/ijms24044243_

Round 1
Reviewer 1 Report
This review is interesting. The only note i suggest to Authors is to clarify better what the affirm in line 280-283. The text is taken literally from the bibliographic entry 44 which dates back to 2017 and which does not specify which markers are. It would be really useful for the reader, and very innovative to know which new markers the Authors refer to.
"Results confirm that establishing the most appropriate patient group for this management requires carefully designed clinical trials. Current prognostic algorithms previously mentioned provide accurate information based on disease and patient characteristics. The next generation of markers might provide personalized prognostic information and predict response to systemic therapies [44].
Author Response
Thank you for your comment. We rephrased the sentence and added some data on biomarkers in the “Current Challenges and Future Perspective” section, accordingly.

Reviewer 2 Report
The risk modelling was not explained authors have cited only one reference for a study which is not convincing.
The overall summary of the trials needs to be given such that readers get the crucial thoughts of the authors.
Author Response
The risk modelling was not explained authors have cited only one reference for a study which is not convincing.
Thank you for the comment. We better explained the risk modelling in the current “Adjuvant rational in RCC” section, and we also better elaborated the last section of the paper.
The overall summary of the trials needs to be given such that readers get the crucial thoughts of the authors.
Thank you for the comment. We improved the “Current Challenges and Future Perspective” section to accordingly.

Reviewer 3 Report
This article focuses on the adjuvant therapy for RCC. The authors have made a detailed analysis of several large related clinical trials. Although most of the results do not have good OS, we can still see the hope of new adjuvant therapy in the future in these experiments. The article is well written, but still has some questions.
1. The paragraphs are too scattered, and some small paragraphs can be merged. for example:In Adjuvant rationale in RCC part, paragraph 1,2,3 can be merged; In Current Challenges and Future Perspective part, paragraphs should be rearranged.
2. In Current Challenges and Future Perspective part, the the content is messy and needs to be rewritten. It is suggested to write in small paragraphs. For example, the main challenge at present is: 1. There is no good classification model to screen out patients in need of treatment. 2. Balance between AEs and effects of drugs. 3. The benefits of DFS and OS are not equal.
3.The resolution of Figure 1 is poor, and clearer pictures need to be uploaded.
4.I don't understand the significance of writing this part(Disease Free Survival as Surrogate for Overall Survival in Patients with Localized RCC). You can add this part to the Current Challenges and Future Perspective section.
Author Response
- The paragraphs are too scattered, and some small paragraphs can be merged. for example:In Adjuvant rationale in RCC part, paragraph 1,2,3 can be merged; In Current Challenges and Future Perspective part, paragraphs should be rearranged.
Thank you for your comment. We merged paragraph 2 and 3, accordingly and better rearranged the last section.
- In Current Challenges and Future Perspective part, the content is messy and needs to be rewritten. It is suggested to write in small paragraphs. For example, the main challenge at present is: 1. There is no good classification model to screen out patients in need of treatment. 2. Balance between AEs and effects of drugs. 3. The benefits of DFS and OS are not equal.
Thank you for your suggestion. We rewrote paragraph reporting the suggested points, accordingly.
3.The resolution of Figure 1 is poor, and clearer pictures need to be uploaded.
Thank you for your comment. We will upload a clearer picture.
4.I don't understand the significance of writing this part(Disease Free Survival as Surrogate for Overall Survival in Patients with Localized RCC). You can add this part to the Current Challenges and Future Perspective section.
Thanks very much for the suggestions. We integrated this paragraph in the current section 2 “adjuvant therapy”.

Round 2
Reviewer 1 Report
I am satisfied with the changes made by the authors. the manuscript, in my opinion, is interesting and in its present form deserves publication. Compliments
Reviewer 3 Report
The revised paragraph structure of the article has improved significantly compared with the previous one, and the modifications made me satisfied. This article will share and discuss the main data and conclusions of adjuvant therapy for renal cell carcinoma after operation, which has certain clinical reference value. it can be accepted.